



# **Seasonal and spatial variability of methane emissions**
# **from a subtropical reservoir in Eastern China**
Yang Le[*], Li Hepeng, Yue Chunlei, Wang Jun
Zhejiang Academy of Forestry, Hangzhou, 310023, China
Corresponding author: [*]E-mail: yangboshi@live.cn



**Abstract:**
Subtropical reservoirs are important source of atmospheric methane ($CH_4$). This study
aims to investigate the spatiotemporal variability of $CH_4$ emission, using the methods
of static floating chambers and bubble traps, from the water surfaces of Xin'anjiang
Reservoir. Seasonal variability showed that $CH_4$ emission from the main reservoir
body was high in autumn and low in spring, with medium values in summer and
winter. The dynamics of $CH_4$ emission was flat from February to June, but fluctuated
dramatically from July to January in the upstream river, which was interrupted by the
bubbles in the second half year. However, $CH_4$ emission was largely influenced by the
streamflow in the downstream river, with a minimum value in February due to an
extreme low streamflow (275 $m^3$ $s^{-1}$). Spatial variability showed the upstream river
had the highest $CH_4$ flux (3.90 ± 7.80 mg $CH_4 \cdot m^{-2} \cdot h^{-1}$), followed by the downstream
river (0.50 ± 0.41 mg $CH_4 \cdot m^{-2} \cdot h^{-1}$), and the main reservoir body stood the last place
(0.01 ± 0.07 mg $CH_4 \cdot m^{-2} \cdot h^{-1}$). Therefore, it was necessary to capture the variation of
$CH_4$ emission from reservoirs in the space and time scales to avoid the error of
estimating the $CH_4$ emission incorrectly.
**Key words:** Spatiotemporal variability; $CH_4$ flux; $CH_4$ emission; Bubble; Xin'anjiang
Reservoir.



## 1. Introduction

Reservoirs are an important type of wetland, which used to be often regarded as clean energy. However, the view was denied by a growing body of researches documenting their role as carbon sources. Deemer et al. (2016) showed that $CH_4$ emissions are responsible for the majority of the radiative forcing from reservoir water surfaces (approximately 80% over the 100-year timescale). The greenhouse gas emission data was limited to 36 Asian reservoirs, among which $CH_4$ emission flux data was available in 3 reservoirs in China, *i.e.*, Three Gorges (Yang et al., 2013; Zhao et al., 2013), Ertan (Zheng et al., 2011), Miyun (Yang et al., 2014). Actually, China had 98,002 dams of different sizes with 142 large-size hydroelectric reservoirs, which did not include the dams under construction or planed now. Thus, more hydroelectric reservoirs distributed in the different geographical regions and climate zones in China should be selected to measure $CH_4$ emission flux to explore the rules of $CH_4$ emission from hydroelectric reservoirs.

Diffusive flux, gas bubble flux, and aquatic vegetation are main pathways for $CH_4$ emission from open water areas in reservoirs (Bastviken et al., 2011). Plant-medium transport is an important $CH_4$ emission pathway in reservoir area with abundant vegetation cover (Bastviken et al., 2011). However, in the no vegetation-distributed areas, ebullition was a dominant way for $CH_4$ emission, while molecular diffusion was a secondary way for $CH_4$ emission from the reservoir water surfaces, although ebullition was found to be episodic (Maeck et al., 2014), because the ebullitive $CH_4$ flux was larger by 1~3 orders of magnitude than the diffusive $CH_4$ flux (Delsontro et al., 2010, 2011). High ebullitive $CH_4$ flux was often observed in the shallow zones, river deltas, and inflow rivers (Delsontro et al., 2010, 2011, 2016), which was influenced by allochthonous organic carbon input and burial (Sobek et al., 2012). Chamber methods were used to measure the $CH_4$ emission flux in the previous studies located in the 3 reservoirs in China, and chamber methods measured the total $CH_4$ emission flux (diffusion plus ebullition) across water-air interface (Yang et al., 2013, 2014; Zheng et al., 2011; Zhao et al., 2013). Probably these previous studies didn't





show the bubble $CH_4$ flux magnitude.

Spatial and temporal variability in $CH_4$ emission are often reported in the reservoirs
(Yang et al., 2013; Zhao et al., 2013; Zheng et al., 2011; Muzenze et al., 2014). The
spatial variability in $CH_4$ emission from reservoirs are caused by the impoundment of
the dams, which changed the hydrological characteristics of the original river.
Upstream and downstream of the dams, outlet of the dam, and inflow rivers to the
reservoirs had distinct $CH_4$ emission levels in a whole reservoir's system (Muzenze et
al., 2014; Kemenes et al., 2007; Abril et al., 2005), because of the hydrological
variables (e.g., water velocity, water depth) (Yang et al., 2013) and dam operation
strategy (Fearnside and Pueyo, 2012). Turning to the temporal variability in $CH_4$
emission, temperature, water column mixing, dissolved oxygen (DO) concentration
and other environmental variables (e.g., retention time, benthic metabolism)
controlled the temporal variability in $CH_4$ emission (Yang et al., 2013; Natchimuthu
et al., 2016; Rodriguez and Casper, 2018). For example, $CH_4$ emission reached the
maximum in the summer and turned to the low levels in the other seasons in the Three
Gorges Reservoir, which was regulated by temperature, DO, and water velocity (Yang
et al., 2013). Temperature regulated the temporal variability of $CH_4$ emission in the 3
lakes (Följesjön, Erssjön, Skottenesjön) of southwest Sweden (Natchimuthu et al.,
2016). Due to the differences in hydrology, water quality, meteorological, and
biological variables, the spatiotemporal variability in $CH_4$ emission should be
explored in the reservoirs, which could understand the differences of $CH_4$ emission in
time and space scales.

Downstream rivers also cannot be ignored because of the degassing fluxes at the
turbines or spillways and high fluxes in the downstream watercourses. Downstream
emission accounted for 50% of total $CH_4$ emissions from the Balbina Reservoir in
Brazil (Kemenes et al., 2007), roughly 30% of total greenhouse gas emissions for the
8 reservoirs in the dry tropical biomes region in Brazil (Ometto et al., 2013), and 10%
of total $CH_4$ emission for Nam Theum 2 Reservoir in Laos (Deshmukh et al., 2016).





Therefore, $CH_4$ emission from the downstream river should be included in a
hydroelectric reservoir.

Two hypothesis are postulated here: (1) the temporal variations in $CH_4$ emission from
water surface are influenced by the temperature, thus a high $CH_4$ emission flux would
be observed in summer and relative low $CH_4$ emission fluxes occurred in other
seasons; (2) upstream and downstream rivers have a great $CH_4$ emission because of
the fast water flow and the low water depth there. The specific objectives in this study
are to investigate the temporal variations in $CH_4$ emission from Xin'anjiang Reservoir,
and upstream and downstream sites are contrasted with those in the reservoir to show
the spatial variations in $CH_4$ emission from the reservoir.

**2. Materials and Methods**
*2.1. Study sites*

**Figure 1.** Dynamics of precipitation, evaporation, air temperature, and water level in the
Xin'anjiang Reservoir region

Xin'anjiang Reservoir (29°28'-29°58'N, 118°42'-118°59'E) is located in the north
subtropical zone, with the mean air temperature of 17.7 °C, the total precipitation of
2015.1 mm, and the total evaporation of 712.9 mm (Figure 1). Xin'anjiang Reservoir
was built in 1959, which has a water area of 567 km², a mean depth of 34 m. The
water storage of the reservoir is about $1.78 \times 10^{10} \, m^3$, the yearly average inflow and
the outflow discharge are $9.4 \times 10^9 \, m^3$ and $9.1 \times 10^9 \, m^3$, respectively, and the water
retention time is about 2 years (Li et al., 2011). Water level fluctuated between 98m to
104m in Xin'anjiang Reservoir in 2015 (Figure 1). The Xin'anjiang Reservoir is
dendritic shape, which consists of northwest lake, northeast lake, southwest lake,
southeast lake, and central lake (Figure 2). Among the 5 sub-lakes, the watercourse of
northwest lake is the most dominant upstream inflow river, which occupy 60~80% of
total surface runoff. Thus, the northwest lake is regarded as the main upstream river of
the Xin'anjiang Reservoir, and the reservoir's main body consisted of northeast lake,



southwest lake, southeast lake, and central lake, and the downstream river is the
watercourse below the Xin'anjiang Dam.

**Figure 2.** The distribution of the sampling transects and sampling sites in the Xin'anjiang
Reservoir

The sampling campaign was conducted in the 4 sub-lakes and the downstream river
(Figure 2). The northwest (NW) lake transect (29°44'03" N, 118°43'04" E) was
located in Jiekou town of Anhui Province, where was the main inflow inlet of
Xin'anjiang Reservoir and had a width of 0.3 km. 3 sampling points (NWP1, NWP2,
NWP3) were chosen from the margin to pelagic zones in the NW transect. The
northeast (NE) lake transect (29°38'44"N, 119°03'03"E) was located in open water
areas of the NE lake near the outlet of a tributary (Jinxianxi). The southwest (SW)
lake transect (29°28'18"N, 118°44'39"E) was located in the open water areas near
Maotoujian Island, where was outlet of the Jiangjia tributary and Fengkou tributary.
The southeast lake (SE) transect (29°28'39"N, 118°45'20"E) was located in the open
water areas between Guihua Island and Mishan Island, where was about 5 km
upstream of the Xin'anjiang Dam. 5 sampling points (from P1 to P5) were chosen
from the margin to pelagic zones in the NE, SW, and SE transects, respectively. In
addition, 4 sampling points were selected in the downstream river below the dam,
with a distance of 0.35 km, 1 km, 4 km, and 7 km away from the Xin'anjiang Dam,
respectively, which was named as DRP1, DRP2, DRP3, and DRP4, respectively.

*2.2. $CH_4$ flux measurements*
In this study, the floating static chambers were used to collect $CH_4$ gas samples from
the surface of Xin'anjiang Reservoir from December 2014 to December 2015.
Monthly measurement was carried out for each sampling site in the morning, and the
measurement lasted for half an hour for each point.    The bubble traps were used to
collect the bubbles in the upstream river from August 2016 to November 2017. The
bubbles were collected once or twice in the NW transect every month except
November, 2016, January and February, 2017, and the collection campaign often




lasted for about 1 day.

The diffusive $CH_4$ emission flux was measured using the static chamber and gas
chromatograph method. The floating static chamber (0.29 $m^2$ for the basal area; 0.117
$m^3$ for the volume) consisted of a plastic box without a cover that was wrapped in
light-reflecting and heatproof materials to prevent temperature variation inside the
chambers; in addition, plastic foam collars were fixed onto opposite sides of the
chamber. The headspace height inside the chamber was about 35 cm. A silicone tube
(0.6 cm and 0.4 cm outer and inner diameters, respectively) was inserted into the
upper central side of the chamber to collect gas samples, and the gas samples were
dried with plexiglass tubes filled with Calcium chloride anhydrous (analytical
reagent), which could remove the moisture in the gas samples and prevent the
biological reactions. Another silicone tube was inserted into the upper corner side of
the chamber to keep the air pressure balanced between the inside and the outside of
the chamber. All measurements were performed in triplicate. The gases in the
headspace of the chamber were collected into air-sampling bags (0.5L; Hedetech,
Dalian, China) four times every 7 min over a 21 min period using a hand-driven pump
(NMP830KNDC; KNF Group, Freiburg, German) (Yang et al., 2013). Once the gas
was collected from the chambers, the gas samples were stored in the air-sampling
bags until analysis in the laboratory. The air-sampling bags made of aluminum can
store the gas samples for 7 days, which does not absorb and react with $CH_4$. The
leakage and memory effects of air-sampling bags have been tested before our
experiments.

The bubble trap consisted of an inverted 30 cm diameter circular funnel fixed with a
closed plastic bottle (volume: 0.56 L) in its narrow neck, and an additional skirt (50
cm diameter circular) was fixed in the large mouth of the funnel to enlarge the bubble
collection range (Wik et al., 2013). Each funnel was stabilized by three equally sized
weights to make sure no tiny bubbles left in the bottles at initial stage. 16 to 26 bubble
traps were fixed in a river-crossing rope with a distance of about 10-15 m between the





two neighbouring bubble traps when the bubbles were sampled. The trapped gas
bubbles would drain the water from the bottles after about 20-40 hours placement.
The left water in the bottles was measured by a graduate to calculate the volume of
trapped gas bubbles. The trapped gas was diluted 1000 times by injecting 1 mL
trapped gas into 1 L or 0.5 L previously $N_2$-filled gas bags, because the $CH_4$
concentration of the trapped gas was too high for the gas chromatograph to reach.

The air-sampling bags were analyzed within 3 days using a gas chromatograph
(Agilent 7890A; Agilent Technologies, Santa Clara, USA) equipped with a flame
ionization detector (FID) and separated with a Teflon column (3 m × 3 mm) packed
with Porpak-Q column (80-100 mesh). The oven, injector, and detector temperatures
were at 70 ℃, 25 ℃, and 200 ℃, respectively. The flow rate of the carrier gas ($N_2$)
was 25 mL·min$^{-1}$, and the flow rate of $H_2$ and the compressed air was set to 40 and 30
mL·min$^{-1}$, respectively. Standard mixed gas ($CH_4$: 1.83 ppm; provided by China
National Research Center for Certified Reference Materials, Beijing) was used to
quantified the $CH_4$ concentration in one of every 10 samples, which kept the
coefficient of variation of the $CH_4$ concentration in the replicated samples below 1%.

The increasing rate of the gas concentration ($dc/dt$) within the static chamber was
calculated as the slope of the linear regression of the gas concentration versus time. It
was suggested that the nonlinear relation between gas flux and time would be better to
determine the steeper initial slope in the chambers. If one plots the time rate of change
of concentration in a closed box, it will be curvilinear, so if measurements were made
at successive time steps, a parabola regression was fit to the data and the slope at time
zero detected (Hutchinson and Livingston, 2001). Thus, the para-curve model was
made preferentially than the linear one. Otherwise, the linear model was accepted.
Acceptance of the results was based upon two criteria: (1) The difference of $CH_4$
concentration between the initial gas sample and ambient air must be within 10% and
(2) the correlation coefficient ($R^2$) had to be > 0.90.



$$F_1 = \rho \times \frac{dc}{dt} \times \frac{273.15}{273.15 + T} \times H \qquad (1)$$
where $F_1$: the diffusive $CH_4$ flux (mg $CH_4 \cdot m^{-2} \cdot h^{-1}$); $\rho$: density of gas under the
standard conditions (0.714 kg$\cdot m^{-3}$ for $CH_4$); H: height of the top of the inverted
chamber to the water surface (0.35 m here); 273.15: absolute temperature at 0 $^o$C; T:
air temperate ($^o$C).
Actually, the static floating chambers can collect both of diffusive and bubble $CH_4$
emission fluxes. Bubbles caused the $CH_4$ concentrations pulses in these chambers.
The average $CH_4$ emission fluxes ($F_a$; mg $CH_4$ m$^{-2}$ h$^{-1}$) in the transects were calculated
by the following equation (2)
$$F_a = \frac{\sum\limits_{n=1}^{n=13}\left[\frac{\sum\limits_{m=1}^{m=5}\left(\frac{\sum\limits_{i=1}^{i=3}F_m}{i}\right)}{m}\right]}{n} \qquad (2)$$
Where, i: the numbers of chambers (3 chambers here); m: the sampling stations in the
transect (NW: 3; NE, SW, SE: 5; DR: 4); n: the total measurement times of $CH_4$
emission during the given time (total times of 13 in 2015, See Table S.1, S.3-6); $F_m$:
the measured $CH_4$ emission flux by the floating chambers.

The mass flux of $CH_4$ via ebullition (bubble $CH_4$ flux) is
$$F_2 = \frac{C_{CH4} \times V \times M}{A_f \times t \times V_m} \qquad (3)$$
Where $F_2$: the ebullitive $CH_4$ flux (mg $CH_4 \cdot m^{-2} \cdot h^{-1}$); $C_{CH4}$: $CH_4$ concentration ($\mu L \cdot L^{-1}$);
V: the accumulated headspace gas volume (L); M: molar weight of $CH_4$ (16.04
g$\cdot mol^{-1}$); $A_f$: the funnel area (0.14 m$^2$); t: the fractional number of hours between
measurement; $V_m$: the molar volume of gas at standard conditions (22.4 L$\cdot mol^{-1}$; gas
samples equilibrated to room temperature before analysis) (Wik et al., 2011).

The ebullition rate (ER; ml m$^{-2}$ h$^{-1}$) reflected the speed of accumulated volume of





bubbles released from the water surface, which was calculated by the following
equation (4).

$$ER = \frac{V}{A_f \times t}$$

235                             (4)

The parameters of V, $A_f$, and t are given in equation (3).

*2.3. Statistical Analysis*
The $CH_4$ flux values were firstly tested by the Kolmogorov-Smirnov test to judge
whether these data satisfied the normal distribution. If not, these $CH_4$ flux data would
be transferred by the trigonometric function or logarithmic function to satisfy the
normal distribution. Then one-way analysis of variance (ANOVA) combined with
Tukey HSD test was used to analyze the seasonal and spatial variability in $CH_4$
emission flux. The data were analyzed using the SPSS (Statistical Product and Service
Solution) 18.0 statistical package.

**3. Results**
*3.1. Seasonal Variations in $CH_4$ Emission*

**Figure 3.** Average $CH_4$ emission from the 3 sampling points in the NW transect of the Jiekou
town between Dec. 2014 to Jan. 2016
Note: NWP1, NWP2, and NWP3 have a distance of about 10 m, 50 m, and 120 m to the south
bank, respectively.

$CH_4$ emission fluxes were measured by the static floating chambers in the upstream
river in 2015, which included the ebullitive and diffusive $CH_4$ emission. The
frequency of bubble occurrence was 16.2% in the NW transect during our
measurement periods (Table S1). The $CH_4$ emission fluxes in the pelagic zones
(NWP2 and NWP3) were significantly higher than those in the margin zone (NWP1),
because no bubbles occurred in NWP1 (Figure 3). $CH_4$ emission from the pelagic
zones was low from February to June, but increased and fluctuated significantly from
July to January, while $CH_4$ emission from the margin zones always kept a low
emission level during the measurement periods (Figure 3).






**Figure 4.** Dynamics of trap bubble flux, ebullition rate, and $CH_4$ concentration in the NW transect.

Note: The X axis of DOY, i.e., days of year, started from $3^{rd}$ August, 2016

Ebullition rates, bubble $CH_4$ emission fluxes, and bubble $CH_4$ concentrations measured using funnel-shaped gas traps in the NW transect, showed a similar seasonal pattern with lower emissions in spring and higher emissions in summer and autumn (Figure 4). Individual measurements ranged from 0 up to 150 mg $CH_4 \cdot m^{-2} \cdot h^{-1}$. The mean bubble flux rate was $22.62 \pm 15.07$ mg $CH_4 \cdot m^{-2} \cdot h^{-1}$ in the NW transect, ranging from 0.31 to 52.27 mg $CH_4 \cdot m^{-2} \cdot h^{-1}$. Measured $CH_4$ concentrations in the collected gas ranged from 7.32 vol. % to 86.03 vol. % with a mean of $59.04 \pm 23.27$ vol. %. The average ebullition rate was $39.93 \pm 24.28$ ml$\cdot m^{-2} \cdot h^{-1}$, ranging from 1.17 to 76.39 ml$\cdot m^{-2} \cdot h^{-1}$. The ebullitive $CH_4$ flux had a significant positive correlated relationship with the ebullition rate ($R^2=0.92$, $p<0.001$, Figure S1), and the bubble $CH_4$ concentration ($R^2= 0.76$, $p<0.001$, Figure S2).

**Figure 5.** Dynamics of diffusive $CH_4$ emission from the 3 transects of reservoir's main body in monthly scale.
Note: The different letters marked in the Fig. 5 indicated that the significant difference was found in the 3 transects during the same sampling periods.

**Figure 6.**   Seasonal variability of $CH_4$ emission from the 3 transects of reservoir's main body
Note: The different letters marked in the Fig. 6 indicated that the significant difference was found in the NE transects among the different seasons.

The dynamic of average diffusive $CH_4$ fluxes fluctuated similarly among the 3 transects in the main body of the Xin'anjiang Reservoir, indicating a fluctuated upwards pattern in 2015, with exception to one sudden peak in $1^{st}$ August (DOY: 213), and one slight peak between $20^{th}$ January (DOY: 20) to $8^{th}$ March (DOY:67) in the SW lake (Figure 5). If $CH_4$ fluxes were analyzed by seasons, seasonal variations in $CH_4$ emission experienced a similar pattern in the NE, SW, and SE transects, which climbed continuously from the minimum in the spring to the maximum in the autumn, but decreased in the winter (Figure 6).





297

**Figure 7.** Dynamics of diffusive $CH_4$ emission from the downstream river.

Note: DRP1, DRP2, DRP3, and DRP4 has a distance of 0.35 km, 1 km, 4 km, and 7 km downstream away from the Xin'anjiang Dam, respectively.

301

The average $CH_4$ flux experienced a similar seasonal variation pattern among the 4 sites in the downstream river (Figure 7): $CH_4$ flux decreased sharply from the maximum value in January to the minimum value in February, and subsequently fluctuated in a relatively small range (Figure 7).

306

*3.2. Spatial Variations in $CH_4$ Emission*

308

**Figure 8.** Average $CH_4$ emission from the different regions in the Xin'anjiang Reservoir.

Note: NW-B, bubble emission from the northwest transect; NW-D: diffusive emission from the northwest transect; NE, northeast lake; SW, southwest lake; SE, southeast lake; DR, downstream river. Different small letters represent the significant difference in average $CH_4$ emission flux among the different transects at the level of p=0.05.

314

The average $CH_4$ emission flux was $3.90 \pm 7.80$ mg $CH_4$ m$^{-2}$ h$^{-1}$ in the NW transect measured by the static floating chambers, including the bubble $CH_4$ flux ($2.73 \pm 2.02$ mg $CH_4$ m$^{-2}$ h$^{-1}$ ) and the diffusive $CH_4$ flux ($1.17 \pm 1.84$ mg $CH_4$ m$^{-2}$ h$^{-1}$; Figure 8). No bubble $CH_4$ emission flux was found in the reservoir main body and the downstream river by the method of the static floating chambers during the whole measurement periods. The average diffusive $CH_4$ emission flux was $0.10 \pm 0.07$ mg $CH_4$ m$^{-2}$ h$^{-1}$ in the main body of the reservoir. Specifically, the average diffusive $CH_4$ emission flux was $0.090 \pm 0.060$ mg $CH_4$ m$^{-2}$ h$^{-1}$, $0.13 \pm 0.086$ mg $CH_4$ m$^{-2}$ h$^{-1}$, $0.079 \pm 0.045$ mg $CH_4$ m$^{-2}$ h$^{-1}$ in the NE, SW, and SE transects, respectively (Figure 8). However, the average diffusive $CH_4$ emission flux increased significantly in the downstream river (DR: $0.50 \pm 0.41$ mg m$^{-2}$ h$^{-1}$; Figure 8). The average diffusive $CH_4$ emission from the main upstream river entrance (*i.e.*, NW transect) and the downstream river exceeded that from the main body of the reservoir (*i.e.*, the NE, SW, and SE transects) by a factor of 11 and 4, respectively (Figure 8).

329





**Figure 9.** Average $CH_4$ emission from the 4 sampling stations in the downstream river.
Note: DRP1, DRP2, DRP3, and DRP4 has a distance of 0.5 km, 1 km, 4 km, and 7 km away from the Xin'anjiang Dam, respectively. Different small letters above the column indicate the significant difference among the 4 sites at the level of p=0.05.

No significant difference was found in $CH_4$ emission from the margin to pelagic zone of the 3 transects in the main body of reservoir. However, the average $CH_4$ emission flux decreased gradually in the downstream river with the distance to the Xin'anjiang Dam, with the maximum in DRP1 ($0.83 \pm 0.43$ mg $CH_4$ $m^{-2}$ $h^{-1}$) and the minimum in DRP4 ($0.33 \pm 0.25$ mg $CH_4$ $m^{-2}$ $h^{-1}$); the average $CH_4$ emission flux in the DRP1 was significantly higher than those of the other 3 sampling points in the downstream river ($p < 0.001$; Figure 9).

## 4. Discussion

### 4.1. Seasonal Variations in $CH_4$ Emission

The dynamics of $CH_4$ emission from the upstream river were influenced by the interference of bubbles, and the peaks of $CH_4$ emission flux in Figure 3 were caused by bubbles (Table S1). In our study, bubbles occurred in the deep zone (>10 m) instead of the shallow zone (<5 m), which was contrary to other studies (Rodriguez and Casper, 2018; Deshmukh et al., 2016). The high ebullitive $CH_4$ emission from the pelagic zone was probably related to the heterogeneity of sediment accumulation (DelSontro et al., 2010, 2011) while no or less sediment accumulation occurred along the margins of the reservoir (Mendonça et al., 2014).

The seasonal variability of $CH_4$ emission from the main body of Xin'anjiang Reservoir denied the hypothesis (1), because the maximum $CH_4$ emission occurred in autumn instead of summer in the 3 transects of the main body, although the significant difference of seasonal variability in $CH_4$ emission was only found in the NE lake (Figure 6). $CH_4$ fluxes had little relationship with air or water temperature after a linear correlated analysis. The explanation to the variability pattern of $CH_4$ emission flux in Figure 6 was probably related with the dynamics of DO concentration in the



water surface. Unfortunately, the DO values were not measured during our sampling
campaigns. But a study on the dynamic distributions of DO in the 6 stations of
Xin'anjiang Reservoir (3 stations overlap with this study) from Jan. 2011 and Dec.
2012 indicated that the maximum DO at surface layer was found in spring and the
minimum value appeared in autumn, because phytoplankton started to breed in the
proper temperature and light conditions at the surface layer in spring, which would
release plenty of oxygen in the water column, while respiration overweight
photosynthesis in autumn (Yin et al., 2014). The variability pattern of DO was
contrary to the dynamics of $CH_4$ flux in Figure 6. $CH_4$ was mineralized to $CO_2$ by
methanotrophic bacteria under aerobic water column (Schubert et al., 2012).

An obvious peak ($0.25 \pm 0.15$ mg $CH_4$ $m^{-2}$ $h^{-1}$) was observed in $1^{st}$ August (DOY: 213)
in the SW lake (Figure 5), and $CH_4$ fluxes in the two margin sampling points (i.e.,
SWP1 and SWP2) were $0.47 \pm 0.11$ mg $CH_4$ $m^{-2}$ $h^{-1}$ and $0.35 \pm 0.081$ mg $CH_4$ $m^{-2}$ $h^{-1}$,
respectively (Table S4), which had a large contribution to the $CH_4$ emission peak. The
high $CH_4$ fluxes from the margin zone were likely attributed to the decomposed
vegetation in the littoral zone when the water level increased to the highest level
(104.4 m) in July (Figure 1). It is worth mentioning that the bank of the SW transect is
gentle and soil slope and the banks of NE and SE transect are steep and rock slope. So
vegetation could grow in the littoral zone of SW transect when the water level was
low enough in spring. Such $CH_4$ emission peaks were also reported in the littoral zone
of Miyun Reservoir and Three Gorges Reservoir (Yang et al., 2012, 2014).

**Figure 10.** The discharge flow in the downstream river below the dam at 9:00 a.m. during the
measurement periods

The downstream $CH_4$ emissions (included the degassing at the turbines) are
proportional to the streamflow in the previous studies (Fearnside and Pueyo, 2012).
The degassing emissions at the turbines of Xing'anjiang Dam were not measured by
the difference in $CH_4$ concentrations at the turbine intake and in the water below the





dam, because about the 500m upstream and downstream of the dam was forbidden to
access to make sure the Xin'anjiang Dam safe. However, $CH_4$ emissions from the 4
sampling points, with the different distances to the dam, were measured 13 times in
2015 (Figures 7). The minimum value ($0.19 \pm 0.11$ mg $CH_4$ $m^{-2}$ $h^{-1}$) appeared in
February, which was likely caused by the low the discharged flow (275 $m^3$ $s^{-1}$) at the
downstream river during the measurement periods (Figure 10). Although the
variability pattern of $CH_4$ emission was not completely consistent with the streamflow
in the downstream river (Figures 7, 10), the streamflow below the dam still account
for 25.3% seasonal variability of $CH_4$ emission in the DRP1 (Figure S3, $p<0.05$,
r=0.50), which was about 500m downstream of the Xin'anjiang Dam.

*4.2. Spatial Variations in $CH_4$ Emission*

**Figure 11.** Schematic diagram of the spatiotemporal variability in $CH_4$ emission from Xin'anjiang
Reservoir
The results were confirmed the hypothesis (2), with a high emission level in upstream
and downstream river, and a low emission level in the main reservoir body (Figure
11). The obviously high $CH_4$ emission from the upstream river was contributed by the
bubbles (Figures 3, 4, Table S2). However, few bubble was trapped in the floating
chambers in the main body of the reservoir and the downstream river below the dam
in 2015 (Figures 6, 7). The $CH_4$ ebullition fluxes in inflow rivers or upstream rivers
were also reported in the many other reservoirs (DelSontro et al., 2011, Musenze et al.,
2014; Beaulieu et al., 2014). Besides the bubble $CH_4$ fluxes, the diffusive $CH_4$ fluxes
contributed to 30% of the total $CH_4$ flux there, and were more than 10 times and 2
times higher than those from the main body and the downstream river, respectively
(Figure 8), which was attributed to the fast water velocity, shallow water depths, and a
large amount of allochthonous carbon input. Water flow was fast in the upstream river
during the heavy rainy days (especially in June), which carried plenty of
allochthonous organic matter constantly. The deepest zone was about 20m in the NW
transect, which was about half to one-third compared with the deepest sampling



points in the 3 transects of the main body. The shallow water depths would reduce the
transport path for small $CH_4$ molecule, and more $CH_4$ would reach the water-air
interface because the less amount of $CH_4$ was oxidized at the oxic layer by the
methanotrophic bacteria (Schubert et al., 2012).

**Table 1.** Previously reported $CH_4$ emission from temperate and subtropical reservoirs


The average $CH_4$ emission fluxes from the upstream river of Xin'anjiang Reservoir
were higher than that of Three Gorges Reservoir, China, Douglas Lake, USA, Nam
Theun 2 Reservoir, Laos, and Eguzon Reservoir, France, but lower than that in
William H. Harsha Lake, USA, Gold Creek and Little Nerang Reservoir, Australia
(Table 1). Diffusive $CH_4$ emission was measured from the upstream rivers of Three
Gorges Reservoir, Douglas Lake, Nam Theun 2 Reservoir, and Eguzon Reservoir,
because no bubble or a few bubbles were observed in the upstream rivers of the 4
reservoirs. A significant high $CH_4$ emission from the upstream river in Xin'anjiang
Reservoir contributed from bubbles, which was similar to the situations in the
upstream rivers of Harsha Lake, Gold Creek, and Little Nerang Reservoir.
Furthermore, The diffusive average $CH_4$ emission from the main body of Xin'anjiang
Reservoir ($0.10 \pm 0.07$ mg $CH_4 \cdot m^{-2} \cdot h^{-1}$) was within the range of $CH_4$ emission level
reported in the other reservoirs in China (mean: $0.22 \pm 0.18$ mg $CH_4 \cdot m^{-2} \cdot h^{-1}$; Li et al.,
2015), but the $CH_4$ emission was 1-2 orders of magnitude lower than that from the
reservoirs in Australia and Laos (William H. Harsha Lake, Gold Creek Reservoir,
Little Narang Reservoir, Nam Leuk and Nam Theun 2 Reservoir), comparable to other
temperate or subtropical reservoirs listed in Table 1, except Douglas Lake and 5 small
reservoirs in Jiangxi Province, China.

Flooded barren soils, dendritic reservoir shape, and aerobic water body probably
caused the relative low $CH_4$ emission from the Xin'anjiang Reservoir. Before the
water storage of the Xin'anjiang Reservoir, strictly clearing activities were done under
the elevation of 70m. The left organic carbon would decompose in the first several



years after impoundment (Abril et al., 2005). After all, Xin'anjiang Reservoir was an
old reservoir with an age of 56-58 years, thus the remaining flooded organic carbon
had little contribution to $CH_4$ emission. Moreover, chlorophyll-a and water depth
controlled the reservoirs $CH_4$ emissions (Deemer et al., 2016). The ranges were in the
range of 1 to 3 μg/L for chlorophyll-a and 10 to 23 μg/L for total phosphorus in the
epilimnion of Xin'anjiang Reservoir, respectively (Li et al. 2011; Yu et al., 2010),
which was an oligotrophic reservoir, according to the classification standard of
nutrition for the tropical/subtropical reservoirs (Cunha et al., 2013). Besides, the
average water depth was about 34 m in the Xin'anjiang Reservoir, and the small $CH_4$
molecules were difficult to pass through such deep path. Furthermore, the Xin'anjiang
Reservoir was dendritic shape, so allochthonous organic carbon mainly deposited in
the sediments of NW lake (Yu et al., 1988, Figure 1), which had little contribution to
$CH_4$ emission from the main reservoir body. In addition, there was no anoxic layer in
Xin'anjiang Reservoir (Zhang et al., 2015), thus the methanotrophic bacteria could
oxidize the dissolve $CH_4$ at the aerobic conditions when they diffused to the
atmosphere (Yang et al., 2014b). All of these above factors combined together lead to
a relative low $CH_4$ emission flux in the reservoir's main body.

A significantly higher $CH_4$ emission was observed in the downstream river than that
in the water surfaces before the dam (Figure 8), which was probably released from the
dissolved $CH_4$ in reservoir's hypolimnions (Abril et al., 2005). Our data set did not
include the dissolved $CH_4$ concentration in different depths before the dam, but
previous related studies reported the dissolved $CH_4$ concentration increased with the
depth before the dam (Abril et al., 2005). The dissolved $CH_4$ would release to the
atmosphere because of the differences in pressure, temperature, and turbulence when
the water passed through the turbines and spillways (Yang et al., 2014b). Strong
turbulence made the dissolved $CH_4$ emission into the atmosphere in the downstream
river below the Xin'anjiang Dam. However, the diffusive $CH_4$ flux dropped with the
distances to the dam, with an obvious higher $CH_4$ flux in the DRP1 (Figure 9), which
was likely related to the decrease of turbulence strength with a distance to the dam



and the explosive release of $CH_4$ gas right after the turbines (degassing). The similar
pattern of $CH_4$ emission was also observed in the downstream rivers of Balbina,
Samuel, Petit-Saut, and Nam Theun 2 reservoirs, and $CH_4$ emission flux in 30 km was
close to the natural rivers nearby (Kemenes et al., 2007; Deshmukh et al., 2016;
Guérin et al., 2006).

**5. Conclusion**
The $CH_4$ fluxes data values obtained in Xin'anjiang Reservoir showed the its different
seasonal variability: $CH_4$ emission from the main reservoir body had a high emission
level in autumn, a low level in spring, and a similar medium levels in summer and
winter; In the main upstream river of the reservoir, $CH_4$ emission was low in the first
half year, but high in the second half year; $CH_4$ emission from the downstream river
was largely influenced by the streamflow below the dam. In the spatial scale, $CH_4$
emission had a high emission level in the upstream river and downstream river, but a
low emission level in the reservoir's main body. A thoroughly investigation should be
carried out in the different reservoir regions for a long-term basis to discover the
spatiotemporal variability in $CH_4$ emission flux in a hydroelectric reservoir system,
which could avoid the error of estimating the $CH_4$ emission incorrectly. The rules on
the temporal and spatial variability in $CH_4$ emission and its potential influencing
variables would be helpful to take proper measures to reduce the greenhouse gases
emissions from the hydroelectric reservoir system in terms of the reservoir's
management.

**Supplementary Materials:**
Figure S1: Positive relationships between the ebullitive $CH_4$ emission and ebullition rates in the
NW transect.
Figure S2. Positive relationship between the bubble $CH_4$ emission and bubble $CH_4$ concentration
in the NW transect.
Figure S3. Positive relationship between the $CH_4$ flux value at DRP1 and streamflow.
Table S1. Complete dataset of the measured $CH_4$ emission fluxes by the floating chambers at the 3




sampling points of NW transect from Dec. 2014 to Jan. 2016.
Table S2. Complete dataset of the measured ebullitive $CH_4$ fluxs, ebullition rates, and $CH_4$
concentrations by the inverted funnels in the 26 sampling stations of the NW transect during Aug.
2016 to Nov. 2017.
Table S3. The measured $CH_4$ emission fluxes by the floating chambers at the 5 sampling points of
NE transect in 2015.
Table S4. Complete dataset of the measured $CH_4$ emission fluxes by the floating chambers at the 5
sampling points of SW transect from Dec. 2014 to Dec. 2015.
Table S5. Complete dataset of the measured $CH_4$ emission fluxes by the floating chambers at the 5
sampling points of SE transect from Jan. 2015 to Jan. 2016.
Table S6. Complete dataset of the measured $CH_4$ emission fluxes by the floating chambers at the 4
sampling points of downstream river from Dec. 2014 to Dec. 2015.

**Acknowledgements:** The study was funded by the National Natural Science
Foundation of China (41303065), the Project of Zhejiang Scientific and Technological
Plan (2015F30001) and Zhejiang Hangzhou Urban Forest Ecosystem Research
Station. We thank for the Xin'anjiang hydropower plant to provide the streamflow
data below the dam. Data presented in this work can be found in the supporting
information.

**Conflicts of Interest:** The authors declare no conflict of interest.

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



# Figures List:





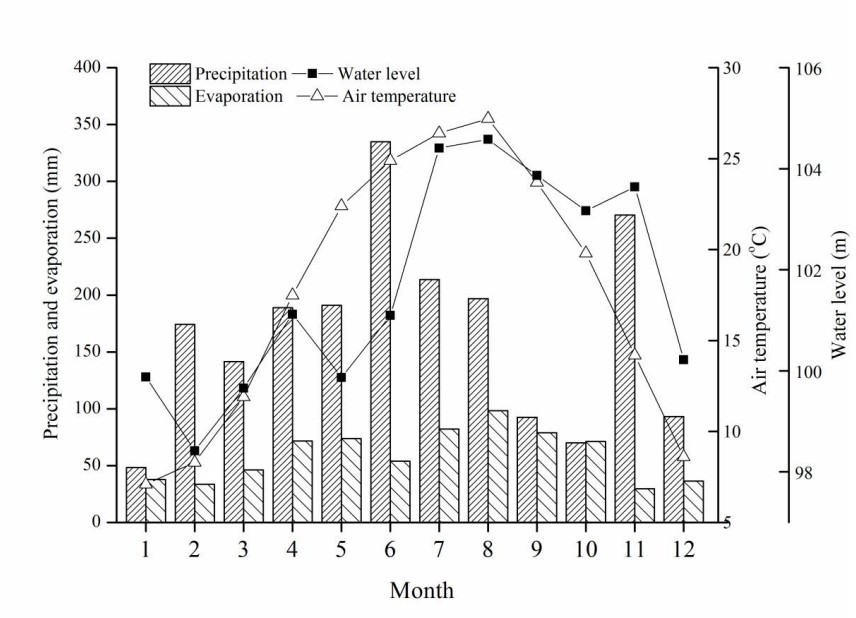


698                                     Figure 1






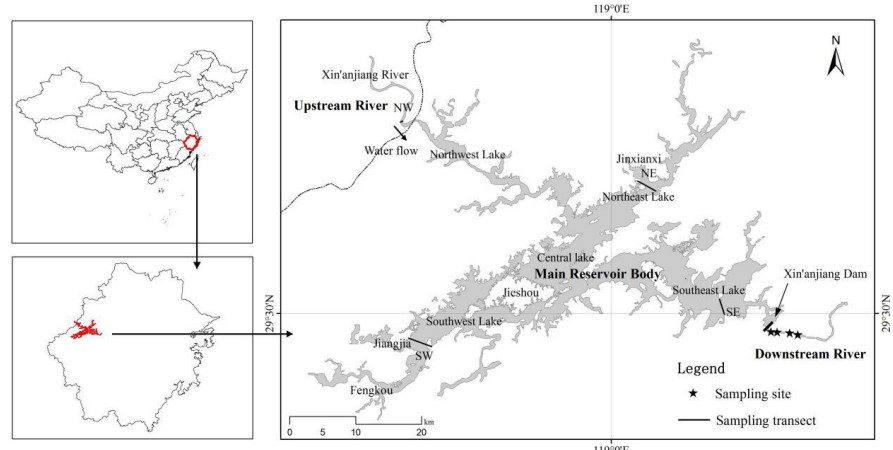


701                                 Figure 2




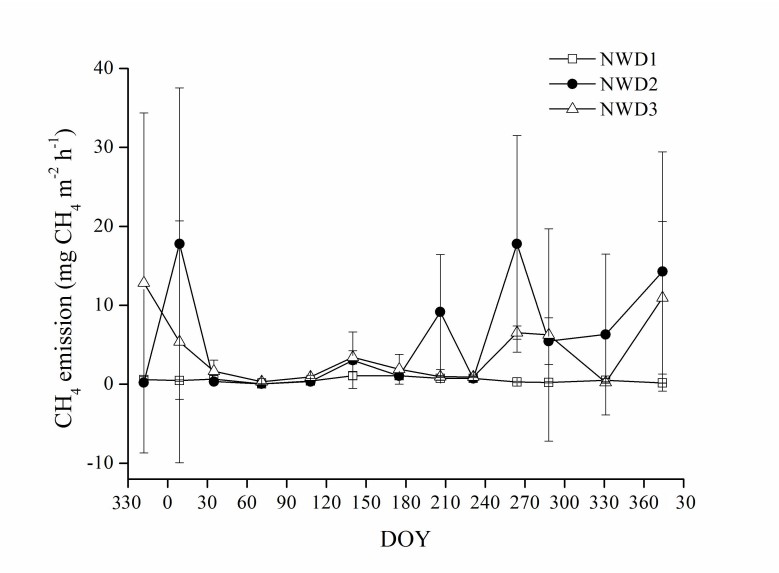


Figure 3






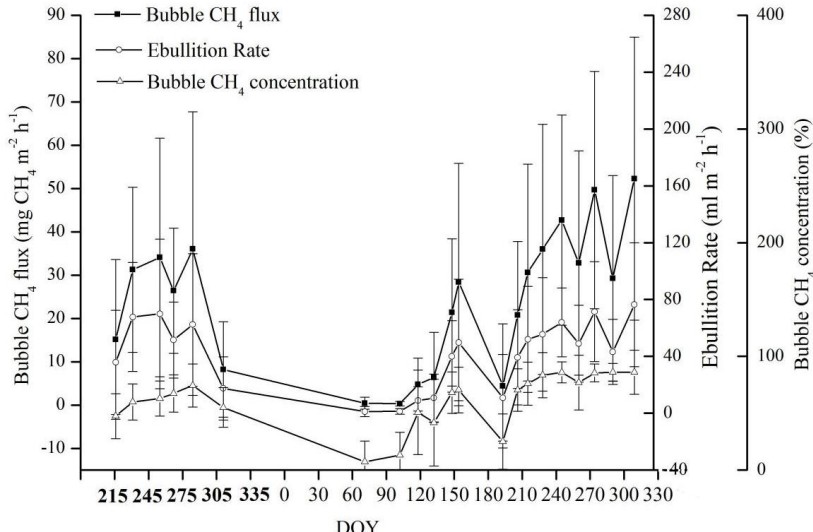

Figure 4




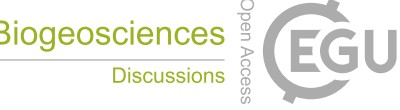



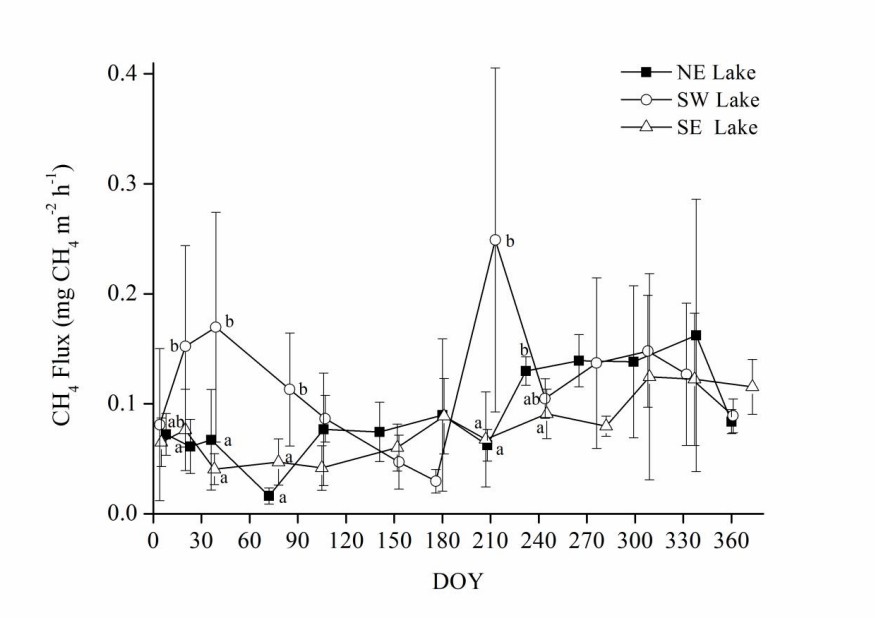


710                                      Figure 5




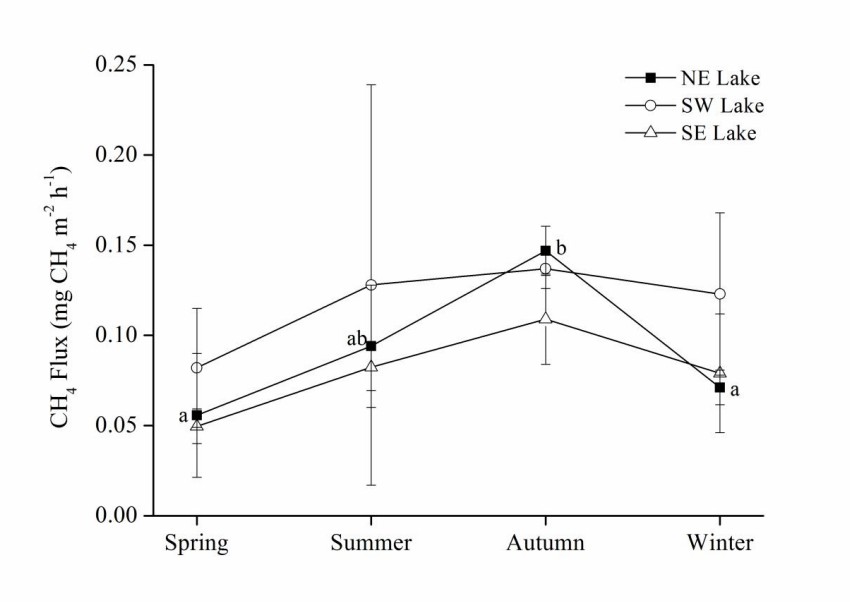


713                                        Figure 6


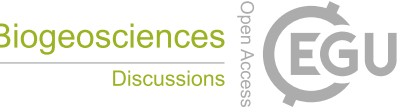


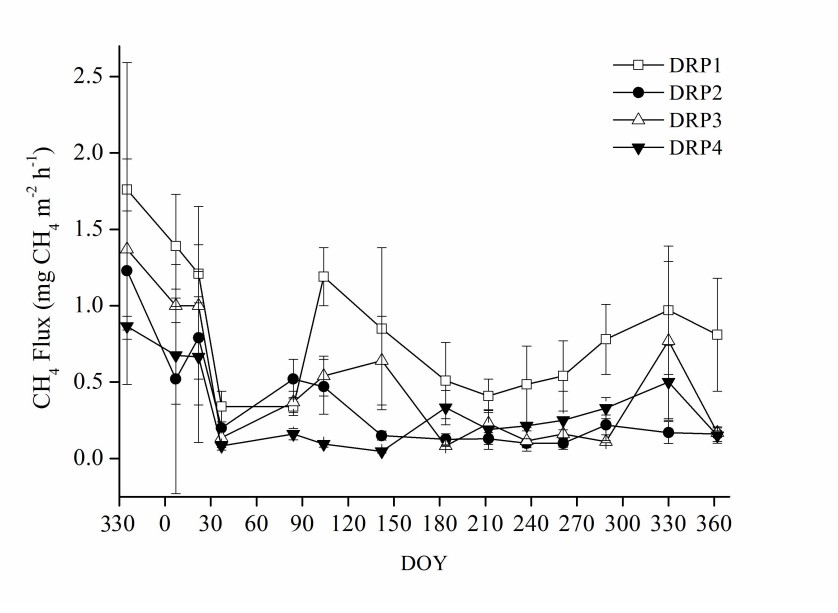


716                                    Figure 7






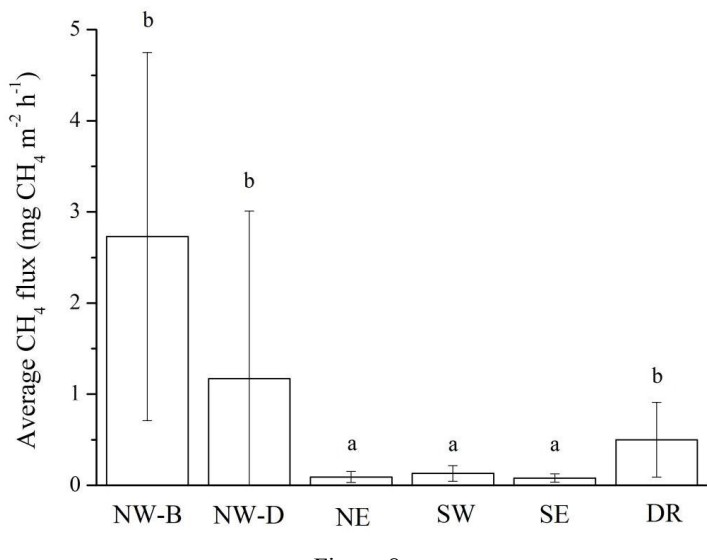


719                          Figure 8




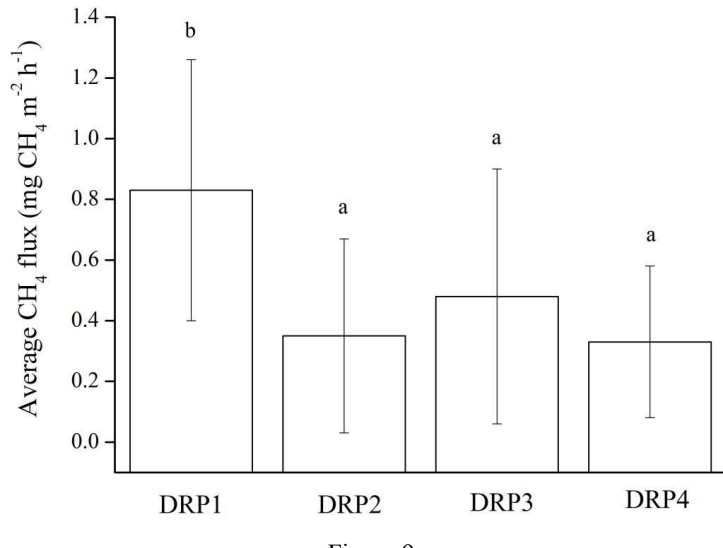


722                     Figure 9






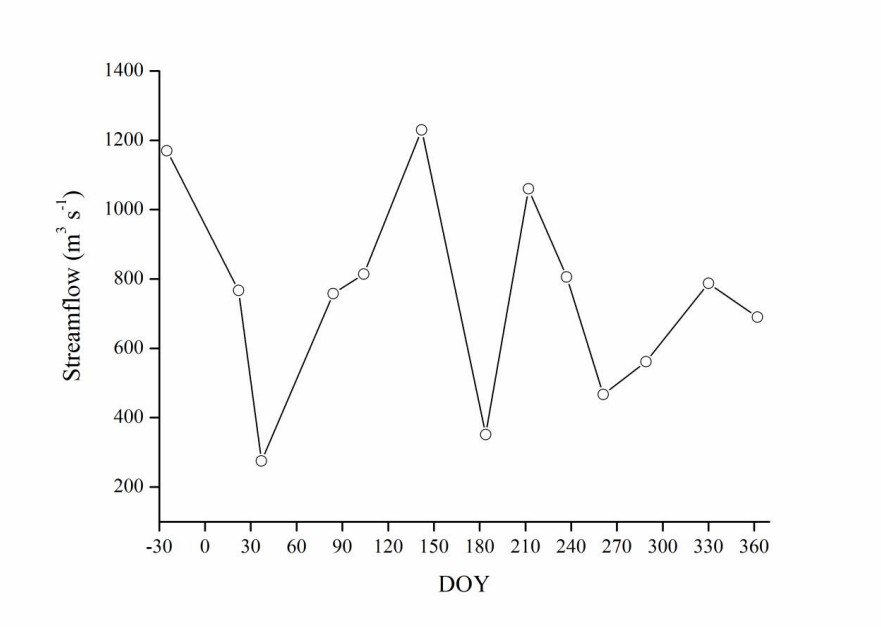


725                                Figure 10




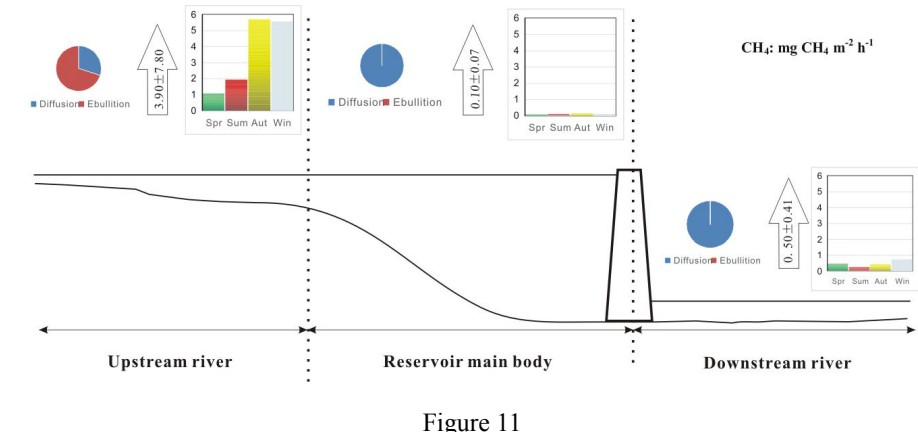


Figure 11







**Table 1.**    **Previously reported CH$_4$ emission from temperate and subtropical reservoirs**

| Country | Reservoir | CH$_4$ Flux (mg CH$_4$ m$^{-2}$ h$^{-1}$) | | | Refs |
|---|---|---|---|---|---|
| | | Upstream river | Open water area | Downstream river | |
| China | Xin'anjiang | 2.73 ± 2.02 (B) | 0.10 ± 0.07 | 0.50 ± 0.41 | 1 |
| | | 1.17 ± 1.84 (D) | | | |
| | Three Gorges | 2.72 ± 1.98 | 0.23 ± 0.40 | 0.26 ± 0.16 | 2,3 |
| | Ertan | | 0.12 ± 0.063 | | 4 |
| | Miyun | | 0.30 ± 0.31 | | 5 |
| | 5 small reservoirs in Jiangxi Province | | 0.013 ± 0.01 | | 6 |
| | 16 small reservoirs in Chongqing | | 0.63 ± 0.89 | | 7 |
| America | William H. Harsha Lake | 130.72 ± 27.50 | 9.77 ± 2.00 | | 8 |
| | Douglas Lake | 0.018 (D) | 0.017 ± 0.012 | | 9 |
| | Eagle Creek | | 0.44 ± 0.73 | | 10 |
| | Six reservoirs in the Western US | | 0.13-0.40 | | 11 |
| Australia | Gold Creek | 172.36 ± 24.72 | 12.35 ± 6.36 | | 12 |
| | Little Nerang Dam | 247.03 ± 254.80 | 6.55 ± 16.83 | | 13 |
| Laos | Nam Leuk | | 1.68 ± 2.68 | | 14 |
| | Nam Ngum | | 0.13 ± 0.13 | | 14 |
| | Nam Theun 2 | 0.9-2.2 | 1.2-2.67 | 8.0 ± 14.7 | 15, 16 |
| France | Eguzon | 0.24 ± 0.56 (B) | 0.4 (0-2.67) | 0.68 ± 0.68 | 17 |
| | | 2.2 ± 3.2 (D) | | | |

Refs: 1. this study; 2. Zhao et al., 2013; 3. Yang et al., 2013; 4. Zheng et al., 2010; 5. Yang et al., 2011; 6. Jiang et
al., 2017; 7.Wang et al., 2017; 8. Beaulieu et al., 2014; 9. Mosher et al., 2015; 10. Jacinthe et al., 2012; 11. Soumis
et al., 2004; 12. Sturm et al., 2014; 13. Grinham et al., 2011; 14. Chanudet et al., 2011; 15. Guérin et al., 2016; 16.
Deshmukh et al., 2016; 17. Descloux et al., 2017. CH$_4$ Flux: B: Bubble emission; D: Diffusive emission.