# Peer review of "Seasonal and spatial variability of methane emissions"

_Biogeosciences, 2018_

## Referee Comment (RC1) · Anonymous Referee #1 · 5 Jun 2018

General remarks

The paper presents measurements of CH4 fluxes between water and atmosphere at a large Chinese reservoir. Monthly measurements at several sites (including the river above and below the reservoir) were performed using floating chambers and bubble traps. The title is somewhat misleading because the main point of the paper is the comparison between the emissions from the (upstream and downstream) rivers with emissions from the reservoir itself. This is comparable to the study about CO2 emissions done by Halbedel and Koschorreck (Biogeosciences, 2013, 10 (11), 7539 – 7551). I recommend to use the comparison of river and reservoir emissions as story-

line in this paper. Hypotheses should be developed along this storyline. Is the message of the paper that the river emits more CH4 than the reservoir? That would mean that constructing a reservoir has the potential to reduce CH4 emissions. The explanation for this somewhat surprising conclusion by the authors is that the deep oxic waterbody slows down emissions by offering more options for methane oxidation. Do you have an idea were the CH4 in the river comes from? Were the emissions in the streams higher because of the CH4 concentration or because of a higher gas transfer coefficient? Was the water released by the dam taken from the hypolimnion? Throughout the manuscript it is often not clear if a particular site is a river, a lake, or a reservoir (e.g. l.117). Lentic and lotic waterbodies, however, are different with respect to GHG emissions. This also has consequences regarding the methods. In a stream you cannot use an anchored chamber (Lorke et al. Biogeosci. 12 (23), 7013 – 7024) while in a lentic waterbody, drifting chambers are problematic because the wind drift might create artificial turbulence. The discussion about the reasons for the observed pattern is often speculative since important data were not measured (e.g. vertical concentration gradients in the reservoir, k600). Considering that the paper presents a rather limited dataset, it is rather long. I think the whole manuscript can be shortened considerably without losing too much information. The method section lacks important information on the field procedures while at the same time somewhat trivial calculations are explained in great detail. Calculation details could be moved to the supplemental material. The language would benefit from a check by a native English speaker.

Detailed remarks

l.13: Remove "Seasonal variability showed that" l.15: "flat" is not the right term here. l.16/17: What does "interrupted by the bubbles" mean? l.22-24: Not a really new finding. l.29: Are reservoirs wetlands? l.29: Replace "used to be often" by "are". l.29: Logic of the sentence is strange. Reservoirs are not Energy. l.36: China has 98002 dams. l.40: Rules is not the correct term here l.44: "plant mediated" instead of "Plant-medium" l.45: Reservoirs typically do not have littoral vegetation. l.46-51: Sentence

rather long. l.57-58: Why probably? Is that not stated in those papers? l.61-63: What is the logic of this sentence? l.77-80: I do not understand what exactly you want to express? l.88: Should be included in a CH4 budget of a hydroelectric reservoir. l.91-94: This is not really a hypothesis because it is long known that temperature has an influence on CH4 emissions. Furthermore, in a deep reservoir the temperature at the bottom is rather constant over the season. The major seasonal difference is the different Corg and maybe O2 supply to the sediment. l.103: Replace "dynamics of" by "monthly". l.104: "... region in 2015." l.107: Mean annual air temperature l.107: "a total" l.108: How was evaporation measured? Is this the evaporation from the water surface? l.109: Remove "which" l.116: Do you mean "contributes" instead of "occupy"? l.117: "inflow" instead of "surface runoff". l.117: Is it a river, a lake or a reservoir? l.113-120: this can be shortened l.125-140: This paragraph can be shortened. Can you also give the water depth at the sampling points (or the elevation of the bottom above sea level (NN))? l.143: Remove "In this study, the" l.145: "at each". What do you mean by morning? Give a range of hours. l.146: Remove article before bubble l.149: Remove the last part of the sentence. l.164: More details about the chamber method are required. Especially: Were the chambers fixed or drifting? 0.5 l sample volume is a lot. Were the measured data corrected for dilution with ambient air during sampling? l.180: At what depth were the traps installed? How long were they deployed? Were the bubble traps installed at all sites? l.185: Remove "to reach". l.187-197: Paragraph is too long l.198-208: Despite the long description it is not fully clear to me which model was used to fit the data. Please provide the equation of the fit. How was the concentration calculated? Provide equation. What is the unit of dc/dt? l.214-217: How was the diffusive flux calculated when ebullition occurred? Equation 2. I wonder whether it makes sense to average over the transects, especially if single sites show extreme values. l.224 add "...flux measured by the bubble traps)" Equation 3: Using equation 3 I get the unit $\mu$g CH4 m-2 h-1, not mg CH4 m-2 h-1. l.251: Add "in the inflowing river" l.268-278: Given the large standard deviation the second digit after the decimal point is not significant and should be removed. l.276: That is no surprise since

the ebullition flux was calculated from the ebullition rate. More instructive would be to show the relation between ebullition flux and CH4 concentration. l.290: What is a "fluctuated upward pattern"? l.302: Remove "average". l.335: This is surprising. Is it because there was no ebullition and the CH4 concentration did not show a lateral gradient? In my experience the CH4 flux from the open water depends strongly on turbulence. Was there a relation between flux and wind speed? l.345-353: This section is not about seasonal variation – move it to 4.2 l.350. What is the definition of the pelagic zone in a river? l.347-348: Does that mean you have to reject hypothesis 2b? l.360-361: How would the O2 concentration at the surface would affect the CH4 flux? I do not belief that. l.398: How would streamflow affect CH4 emissions? What is the mechanism behind this dependency? l.417: How fast was the water flow? If you used a fixed chamber, the high flux was probably an artefact because the chamber would create artificial turbulence which accelerates the flux (as shown by Lorke et al.). l.422: Vertical transport of CH4 in the water column is usually limited by the slow diffusion through the thermocline. Thus, the thickness of the hypolimnion or epilimnion is not so important. l.428-445: Paragraph too long. L.469: Where the emissions from the river high because of the CH4 concentration or because of a high k600? l.474: You cannot simply transfer the results from another, completely different, reservoir. l.476-478: Sentence sounds strange. l.495-498: I thought that exactly this was the purpose of this study. Fig.4: how can the CH4 bubble concentration be higher than 100%? Fig.6: Move to supplemental material

---

## Referee Comment (RC2) · Anonymous Referee #2 · 21 Jun 2018

Reported greenhouse gas fluxes from reservoirs located in subtropical zone are still insufficient to understand regional carbon cycling. The manuscript titled "Seasonal and spatial variability of methane emissions from a subtropical reservoir in Eastern China" presents methane effluxes from a reservoir with an age of about 60 years, which is helpful to understand the topic. Although I would give it a positive response, a major revision should be made for being reviewed again. I hope some key environmental factors (wind speed, water velocity, water temperature, air temperature, water depth of each site in different periods, and so on ) should be introduced to elucidate their viewpoints instead of guessing.

[Figure]

Specified comments: 1. Lines 125-137: Water depths should be presented for each site.

2. Lines: 201-208 I don't think a para-curve model is suitable to fit gas concentrations in a chamber over time. The cited paper of Hutchinson and Livingston is about measuring gas exchange between soil and atmosphere. Pls see the following references. Xiao, S., Wang, C., Wilkinson, R.J., Liu, D., Zhang, C., Xu, W., Yang, Z., Wang, Y., Lei, D., 2016. Theoretical model for diffusive greenhouse gas fluxes estimation across water-air interfaces measured with the static floating chamber method. Atmospheric Environment 137, 45-52. Tanka P.Kandel, Poul ErikLærke, Lars Elsgaard. Effect of chamber enclosure time on soil respiration flux: A comparison of linear and non-linear flux calculation methods. Atmospheric Environment 141, 245-254

3. Line 218: Are there some mistakes in the equation (2)? The denominators m, n and i may be 5, 13 and 3 respectively?

4. Lines 126-128: I think the natural physical geographical characteristics described here is important. I think the changing hydrological situation may influence gas fluxes in the NW Lake more or less. For example, can you correlate it with Lines of 315-328?

5. Lines 354-370 This paragraph tries to explain "The seasonal variability of CH4 emission from the main body of Xin'anjiang Reservoir". However, I think they are feeble. I hope more environmental factors should be taken into considerations, such as wind speed, difference between the air temperature and water temperature, and so on.

6. Lines 372-382: Reasons presented here for the high value of methane flux in the SW lake on August 1 are also weak. Figure 5 shows the methane flux in Feb. is also high, but the water level during the mon was low.

7. English language needs a bigger improvement. For example: 1). Line 233: "from" should be "across"

2. Line 271: "Individual measurements" should be "The flux of individual measurements"

---

## Author Comment (AC1) · 15 Jul 2018

The paper presents measurements of CH4 fluxes between water and atmosphere at a large Chinese reservoir. Monthly measurements at several sites (including the river above and below the reservoir) were performed using floating chambers and bubble traps. The title is somewhat misleading because the main point of the paper is the comparison between the emissions from the (upstream and downstream) rivers with emissions from the reservoir itself. This is comparable to the study about CO2 emissions done by Halbedel and Koschorreck (Biogeosciences, 2013, 10 (11), 7539-7551). I recommend to use the comparison of river and reservoir emissions as storyline in this

paper. Hypotheses should be developed along this storyline. Response: According to the reviewer's suggestions, the title of manuscript is changed into "Contrasting methane emissions from upstream and downstream rivers of an associated subtropical reservoir in eastern China" (lines 1-3). we referred the literature of Halbedel and Koschorreck (2013), the comparison of CH4 emissions from (upstream and downstream) rivers and the associated reservoir was used as storyline in the revised manuscript, and the related hypotheses also proposed in lines 98-100.

Is the message of the paper that the river emits more CH4 than the reservoir? That would mean that constructing a reservoir has the potential to reduce CH4 emissions. The explanation of this somewhat surprising conclusion by the authors is that the deep oxic waterbody slows down emissions by offering more options for methane oxidation. Response: According to our data, the upstream and downstream rivers emit much more CH4 than the reservoir surfaces (lines 19-21), so reservoir construction indeed has the potential to reduce CH4 emission if compared with CH4 emission from the original lotic river (lines 22-24). On the other hand, constructing a reservoir flooded large quantities of soils in the watershed near the original rivers, which would transfer a CH4 sink into a CH4 source in the new flooded areas. The original hydrology conditions of natural river are changed after impoundment of reservoir, and the increased water depth, decreased water velocity, and thermal stratification in the large, deep reservoir like Xin'anjiang would had a potential influence to decline CH4 emission compared with the upstream inflow rivers. Turning to the downstream river, the dissolved CH4 in the hypolimnion before the dam tend to release to the atmosphere in the downstream river because of the differences in pressure and temperature. So the downstream river would has more CH4 emission than the reservoir surfaces.

Do you have an idea were the CH4 in the river comes from? Were the emissions in the streams higher because of the CH4 construction or because of a higher gas transfer coefficient? Was the water released by the dam taken from the hypolimnion? Response: According to the thin boundary layer model, CH4 flux (F) was determined by the gas

transfer coefficient (k) and the air-water CH4 concentration gradient ($\Delta C$=Cwater - Csat), i.e., F=k×$\Delta C$. Thus, enhancement of them can increase the CH4 flux at the air-water interface. However, the dissolved CH4 concentrations in the water were not measured in our experiments, so we lack of the direct evidences of the impact of k, $\Delta C$ on F. The viewpoint stands from the CH4 emission at the air-water interface, actually, the diffusive CH4 flux was determined by the CH4 production in the sediments and the CH4 consumption (i.e, oxidation) during the vertical transport. The high CH4 flux in the upstream river was supposed to derived from the high rate of CH4 production in the sediments, because soil erosion and other man-made pollution enhance the organic carbon input in the upstream river (lines 361-363). The standardized Schmidt number of 600 (k600) was often reported to be positively correlated with wind speed. However, the average wind speeds in the upstream and the downstream rivers (3.39 m/s, 3.85 m/s, respectively) are close to those in NE, SE, and SW (4.43, 2.92, and 3.06 m/s respectively). So the spatial variability of CH4 flux probably had a weak relationship with wind speeds (Table 2). But water velocity, similar to wind speeds, also bring turbulent mixing on the air-water interface in a large river like Xin'anjiang River, which could see the literature (Beaulieu J, Shuster W, Rebholz J, Controls on gas transfer velocities in a large river, Journal of Geophysical Research, 2015). Xin'anjiang Reservoir is clam during the no or weak wind periods, because the main body of reservoir likes a huge clam lake with almost no water currents, but the (upstream and downstream) rivers had obviously water currents. The water velocity in the upstream river is high in June, while relatively low in other months, but in the downstream river, the water velocity was controlled by the dam operation, with a high water velocity during the daytime and low during the night.

Throughout the manuscript it is often not clear if a particular site is a river, a lake, or a reservoir (e.g. l.117). Lentic and lotic waterbodies, however, are different with respect to GHG emissions. This also has consequences regarding the methods. In a stream you cannot use an anchored chamber (Lorke et al. Biogeosci. 12 (23), 7013-7024) while in a lentic waterbody, drifting chambers are problematic because the wind drift

might create artificial turbulence. Response: The particular site (Jiekou) in NW lake is located in a river. And the water velocity was fast in June, while kept a low level in other months. During our campaign, we avoid the flooded periods in June, and the chambers were tethered to a drifting boat (line ??). Although strong wind speeds create artificial turbulence, wind speeds was relatively low (<3 m s-1) most time when the gas sampling was collected in our study.

The discussion about the reasons for the observed pattern is often speculative since important data were not measured (e.g. vertical concentration gradients in the reservoir, k600). Response: Just as the reviewer's comments, some important data was not measured in our experiments, and it is difficult to discuss the results in the discussion section. Despite some uncertainties, the mechanisms will be studied further, i.e. the effect of thermal stratification on CH4 emission from reservoir surfaces.

Considering that the paper presents a rather limited dataset, it is rather long. I think the whole manuscript can be shorten considerably without lossing too much information. Response: The unimportant information was deleted from manuscript, according to the following suggestions (lines 116-120, 124-134, 177-184).

The method section lacks important information on the field procedures while at the same time somewhat trivial calculations are explained in great detail. Calculation details could be moved to the supplemental material. Response: According to the reviewer's suggestions, the important information was supplemented (lines 158-159) and three equations (i.e., Eq. S1, Eq. S2, Eq. S3) were deleted from the revised manuscript, which was moved into the supplemental material.

The language would be benefit from a check by a native English speaker. Response: The language was polished by a professional company.

Detailed remarks l.13: Remove "Seasonal variability showed that" Response: As the reviewer's suggestion, "Seasonal variability showed that" was deleted from the revised manuscript.

l.15: "flat" is not the right term here. Response: "was flat" was replaced by "low" (line 15).

l.16/17: What does "interrupted by the bubbles" mean? Response: "interrupted by the bubbles" was replaced by "due to bubble activity" (line 16), because the occurrence of bubbles had a large impact on the fluctuation of CH4 emission dynamics in the second half of year.

l. 22-24: Not a really new finding. Response: The original sentence was deleted and a new one instead in lines 24-26.

l.29: Are reservoirs wetlands? l.29: Replace "used to be often" by "are". l.29: Logic of the sentence is strange. Reservoirs are not Energy. Response: To avoid unnecessary mistakes, the first sentence in the introduction was changed into "Hydropower has previously been regarded as a clean source of energy; however, this view is challenged by a growing body of research that has reported are carbon sources" (lines 31-33).

l.36: China has 98002 dams. Response: China has over 98, 000 dams (line 39).

l.40: Rules is not correct term here. Response: "changing pattern" was used here (Line 40).

l.41: "plant mediated" instead of "Plant-medium" Response: "Plant-medium" was changed into "plant-mediated" in line 46.

l.45 Reservoirs typically do not have littoral vegetation. Response: Different reservoirs had distinct distribution of littoral vegetation. In some reservoirs, emergent plant, e.g., reed, was distributed in the littoral zone, which has a large contribution to methane emission.

l.46-51 Sentence rather long. Response: The original sentence was divided into 2 short sentences (lines 47-51).

l.57-58 Why probably? Is that not stated in those papers? Response: Bubbles were

not measured by the inverted funnels (bubble traps) in those three reservoirs, and only floating chambers was used to collect gas samples to measure the diffusive CH4 emission, occasionally bubbles were captured in the chambers (lines 54-58).

l.61-63 What is the logic of this sentence? Response: The sentences are changed in lines 61-65, which show the consequences of river impoundment and their further impact on spatial variability of CH4 emissions.

l.77-80: I do not understand what exactly you want to express? Response: The sentence is a summary one to show the environment factors have an impact on CH4 emission from reservoirs, so the spatiotemporal variability should be stressed to avoid the error of estimating the CH4 emission incorrectly (lines 77-80).

l.88: Should be included in a CH4 budget of a hydroelectric reservoir. Response: CH4 emission from the downstream river should be included in the CH4 budget of a hydroelectric reservoir system. The last sentence of the paragraph was deleted in the revised manuscript, because it is implicit in the preceding text.

l.91-94: This is not really a hypothesis because it is long known that temperature has an influence on CH4 emissions. Furthermore, in a deep reservoir the temperature at the bottom is rather constant over the season. The major seasonal difference is the different Corg and maybe O2 supply to the sediment. Response: According to the reviewer's suggestions, the hypothesis was deleted from the revised manuscript.

l.103: Replace "dynamics of " by "monthly". l.104: ". . . region in 2015." Response: Accepted in the revised manuscript (lines 113-114).

l.107: Mean annual air temperature Response: The expression was revised according to the reviewer's suggestion (line 105).

l.107: "a total" Response: Deleted. Because the express was changed in lines 105-106.

l.108: How was evaporation measured? Is this the evaporation from the water surface?

Response: The evaporation data was provided by a local meteorological station. I don't
know how to measure evaporation.

l.109: Remove "which". Response: Deleted the word in line 107.

l.116: Do you mean "contributes" instead of "occupy"? Response: "contributes" was
accepted in the revised manuscript (line 118).

l.117: "inflow" instead of "surface runoff". Response: Accepted in line 119.

l.117: Is it a river, a lake or a reservoir? Response: Actually, the NW lake is a river,
because most water of reservoir derived from the NW lake.

l.113-120: this can be shortened Response: These sentences were shortened in lines
116-120.

l.125-140: This paragraph can be shorten. Can you also give the water depth at the
sampling points (or the elevation of the bottom mean by morning? Give a range of
hours) Response: The paragraph has been shorten (lines 124-133), and Table 1 was
given to show the depths of sampling points (line 135).

l.146: Remove article before bubble. Response: The article was deleted.

l.149: Remove the last part of the sentence Response: The last part of the sen-
tence was removed (line 144). The measurement of wind speed and temperatures
was added in lines 144-146.

l.164: More details about the chamber method are required. Especially: Were the
chambers fixed or drifting? 0.5 l sample volume is a lot. Were the measured data
corrected for dilution with ambient air during sampling? Response: Three chambers
were deployed in the water surface near the sampling points, but the chambers were
tethered to a boat, which was drifted in the reservoir surface (lines 158-159). In addi-
tion, the total volume of a chamber is about 117L (line 150), and the maximal volume
of a sampling bag is 0.5 L. However, only about 0.2-0.3L gas was collected into the

bags once, so the total collected gas was about 1L, which was less than 1% of the total volume, so such influence could be ignored.

l.180: At what depth were the traps installed? How long were they developed? Were the bubble traps installed at all sites? Response: Water depth ranged between 5m and 25m (line 168). The traps were deploy about 24 h (line 174). The bubble traps were only deployed in the NW transect, because almost no bubble was captured by the static chambers in the other transects.

l.185. Remove "to reach". Response: Accepted as the reviewer's suggestion (line 177).

l.187-197: Paragraph is too long Response: Some trivial information was deleted in the paragraph (lines 177-184).

l.198-208: Despite the long description it is not fully clear to me which model was used to fit the data. Please provide the equation of the fit. How was the concentration calculated? Provide equation. What is the unit of dc/dt? Response: CH4 concentrations in the sampling bags were measured by a gas chromatograph instead of calculation, and its unit is ppm. The CH4 flux (equation 1) is determined from the change of the gas concentration (△c) in the chamber headspace over time (△t). According to the suggestions by the reviewer #2, the linear regression is used to fit the measured CH4 concentration data instead of para-curve model (line 187). Actually, the equation of the fit is d△c/d△t, i.e., the increased CH4 concentration at the given time, and there is no unit for dc/dt.

l.214-217: How was the diffusive flux calculated when ebullition occurred? Equation 2. I wonder whether it makes sense to average over the transects, especially if single sites show extreme values. Response: From our data sets by the floating chambers, bubbles occurred in the NW transect. Bubbles were considered to be trapped in the chambers, when the CH4 concentrations increased significantly and abruptly in the chambers. The ebullitive CH4 flux in the bubble-trapped chamber was close to 2 magnitude higher than the diffusive CH4 flux of the bubble-free chamber nearby. Thus, the diffusive flux was ignored when the bubble enter the chambers. The extreme values of CH4 flux mainly caused by the bubbles, and the ebullitive CH4 fluxes were calculated separately from the diffusive ones. In the NW transect, the frequency of bubble occurrence was 16.2% during our measurement periods, and the calculation of average CH4 emission flux in the NW transect was calculated by the following equation (Eq. S2): Faverage=16.2%×Febullition+83.8%×Fdiffusion In the other transects with almost no bubble, the average CH4 flux was calculated by the geometric mean of all the measured CH4 fluxes by the 3 chambers in all sampling sites during the measurement periods.

l.224 add "... flux measured by the bubble traps)" Response: The phrase was added in lines 199-200.

Equation 3: Using equation 3 I get the unit $\mu$g CH4 m-2 h-1, not mg CH4 m-2 h-1. Response: The unit of F2 in Eq. 2 is mg CH4 m-2 h-1, the transfer coefficient (1/1000) was added in equation 3 (line 202). However, there is no error in the data in the results, because the bubble gas samples were diluted 1000 times by pure N2, which offset the transfer coefficient (1/1000).

l.251: Add "in the flowing river". Response: The caption of Fig. 3 was changed in lines 223-224.

l.268-278: Given the large standard deviation the second digit after the decimal point is not significant and should be removed. Response: The suggestion is accepted in the results section (lines 226-233).

l.276: That is no surprise since the ebullition flux was calculated from the ebullition rate. More instructive would be to show the relation between ebullition flux and CH4 concentration. Response: The Eq. 2 and Eq. S3 are as follows: (4) (5) Actually, , here C is a constant (7.16×10-4), i.e., That is to say, the ebullitive flux (F2) is determined by the ebullition rate (ER) and the CH4 concentrations (CCH4), so we easily found the ebullitive flux was positively correlated with ER and CCH4 (Fig. S1a). The measured CH4 concentrations in the bubble traps and its corresponding CH4 fluxes were positively correlated in Fig. S1b (model: y=0.52x-7.20, R2=0.36, p<0.001, n=456), although with low R2 compared with the regression model between CH4 flux and ebullition rate (y=0.61x-1.10, R2=0.87, p<0.001, n=456).

l.290: What is a "fluctuated upward pattern"? Response: The phase was deleted in the revised manuscript, and the expression was changed in lines 238-239.

l.302: Remove "average". Response: The expression of the sentence was changed in lines 250-251.

l.335: This is surprising. Is it because there was no ebullition and the CH4 concentration did not show a lateral gradient? In my experience the CH4 flux from the open water depends strongly on turbulence. Was there a relation between flux and wind speed? Response: There was no significant difference in the lateral variability of CH4 emission in the NE, SW, and SE transects after one-way ANOVA test (Fig. S3c, d, e). In the 3 transects of the main body, there was no bubble trapped in our chambers in 2015. According to the reviewer's suggestions, CH4 fluxes and wind speed was correlated at each transect, and the results is listed in Table 2 of the revised manuscript. Although the correlations are significant (p<0.05), wind speed has a small account for the variability of CH4 flux (a low value of R2). However, some high CH4 fluxes were indeed caused by the turbulence of strong wind (> 10 m s-1), e.g., the high CH4 fluxes in SW transect on 8 February (SWP2: 0.23 mg m-2 h-1, SWP4: 0.20 mg m-2 h-1).

l. 345-353: This section is not about seasonal variation — move it to 4.2 Response: The section was moved to 4.2 (lines 345-351).

l.350: What is the definition of the pelagic zone in a river? Response: Pelagic zone here is the deep water zone, e.g. the central zone.

l.347-348: Does that mean you have to reject hypothesis 2b? Response: From our

results, bubbles have an obviously contribution to the high CH4 emission from the upstream river, however, it don't mean the shallow water depth and fast water velocity are not important for CH4 emission there, because many environmental factors have interaction effects. So we don't rejected the hypothesis.

l.360-361: How would the O2 concentration at the surface would effect the CH4 flux? I do not belief that. Response: According to reviewer's suggestion, the related content was deleted from revised manuscript. Thermal and dissolved oxygen stratification become weak in the second half of year, which probably related with the increased diffusive CH4 flux, because Vertical transport of CH4 in the water column is usually limited by the slow diffusion through the thermocline (or oxycline).

l.398: How would streamflow affect CH4 emissions? What is the mechanism behind this dependency? Response: Streamflow at the outlet of dam had close relationship with water velocity in the downstream river, and a high water current velocity would enhance gas transfer velocity (k) at the air-water interface (lines 85-87) (Beaulieu et al., Controls on gas transfer velocities in a large river, Journal of Geophysical Research, 2012). In addition, Fearnside and Pueyo (2012) considered that downstream emissions are proportional to the streamflow after comparing the results between Tucuruí and Balbina in Brazil.

l.417: How fast was the water flow? If you used a fixed chamber, the high flux was probably an artefact because the chamber would create artificial turbulence which accelerates the flux (as shown by Lorke et al.). Response: Water flow was not measured in our experiments. The monitored transect (NW transect) is the main upstream inlet of the reservoir, with a fast water inflow in June because of the heavy rains in its upstream watershed. We collect the gas samples during the intermission of floods (24 June). The chambers were freely drifting while followed with our boat, the measured CH4 flux in such conditions was more reliable than the anchored chambers (Lorke et al., 2015), although there is turbulence more or less.
l.422: Vertical transport of CH4 in the water column is usually limited by the slow diffusion through the thermocline. Thus, the thickness of the hypolimnion or epilimnion is not so important. Response: According to the reviewer's suggestion, the relevant content was deleted from the revised manuscript.

l.428-445: Paragraph too long Response: The paragraph was shortened in lines 366-373.

l.469: Where the emissions from the river high because of the CH4 concentration or because of a high k600? Response: We lacked of direct evidences to explain the high CH4 emissions from the downstream river, however, plenty of CH4 emitted in the downstream river when the hypolimnion water passed through turbines in other dams (Kemenes et al., 2007; Deshmukh et al., 2016), because of the differences in temperature and pressure promote the high dissolved hypolimnion CH4 concentrations release in the downstream river (lines 83-85, 401-405). In addition, the remainder of the paragraph are deleted from the revised manuscript, because CH4 flux don't show the decrease trend with the distance to the dam after the amendment of CH4 fluxes in Eq. 1 (Figure S3 a).

l. 474: You cannot simply transfer the results from another, completely different, reservoir. Response: According to the reviewer's suggestion, the sentence was deleted from our revised manuscript.

l.476-478: Sentence sounds strange. Response: It contains some uncertainty on the strong turbulence on CH4 emission from the downstream river, so the sentence was deleted from original manuscript.

l.495-498: I thought that exactly this was the purpose of the study. Response: The sentences was moved to the abstract (lines 24-26).

Fig. 4: How can the CH4 bubble concentration be higher than 100%? Response: The CH4 bubble concentration values did not exceed 100%, and the y coordinate axis

ranged from 0 to 400% to avoid to overlap with other data. To avoid the misunderstanding proposal by the reviewer, the range of y coordinate axis of bubble CH4 concentration changes from -30% to 120% to show the complete error bars in the revised manuscript (Figure 4).

Fig. 6: Move to supplemental material. Response: As the reviewer's suggestion, the original Fig. 6 was moved into supplemental material (Figure S2).

Please also note the supplement to this comment:
https://www.biogeosciences-discuss.net/bg-2018-195/bg-2018-195-AC1-supplement.pdf

———————————————————

[Figure]

[Figure]

**Fig. 1.** Monthly precipitation, evaporation, air temperature, and water level in the Xin'anjiang Reservoir in 2015.

[Figure]

**Fig. 2.** Location of transects and sampling points.

[Figure]

**Fig. 3.** Mean CH4 emissions from the upstream river of the reservoir between December 2014 to January 2016.

[Figure]

**Fig. 4.** Mean ebullition rate, bubble CH4 emission flux, and bubble CH4 concentration recorded from the inflow river.

[Figure]

**Fig. 5.** Mean monthly diffusive CH4 emission from the areas of the reservoir.

[Figure]

**Fig. 6.** Mean diffusive CH4 emissions from different distances downstream of the reservoir.

[Figure]

**Fig. 7.** Mean CH4 emissions from the reservoir and the upstream and downstream rivers.

[Figure]

**Fig. 8.** Discharge flow at 9:00 hrs in the downstream river below the dam during the measurement period.

**Supplement:**

Supporting Information for

**Methane emissions from the upstream and downstream rivers and their intermediate reservoir in Eastern China**

Yang Le, Li Hepeng, Yue Chunlei, Wang Jun

Zhejiang Academy of Forestry, Hangzhou, 310023, China

**Contents of this file**

Average flux in CH$_4$ emissions (F$_a$; mg CH$_4$ m$^{-2}$ h$^{-1}$) from the transects was calculated as (Eq. S1):

$$F_a = \dfrac{\displaystyle\sum_{n=1}^{n=13}\left[\dfrac{\displaystyle\sum_{m=1}^{m=5}\left(\dfrac{\displaystyle\sum_{i=1}^{i=3} F_m}{i}\right)}{m}\right]}{n} \qquad\qquad (S1)$$

where, $i$ is numbers of chambers; $m$ is the number of sampling points within a transect; $n$ is the number of times CH$_4$ emissions were measured during a given period (Table S.1, S.3-6); and, $F_m$ is CH$_4$ emission flux measured by the floating chambers.

Since static floating chambers collect diffusive and bubble CH$_4$ emissions, pulses in CH$_4$ concentrations were driven by bubbles: therefore, average flux in CH$_4$ emissions were calculated as the sum of the frequency of diffusive and ebullitve CH$_4$ emissions ($F_a$) (Eq. S2):

$$F_a = f \times F_{ebullition} + (1-f) \times F_{diffusion} \qquad\qquad (S2)$$

where, $f$ is frequency of bubble occurrence; $F_{ebullition}$ is geometri mean of CH$_4$ fluxes in chambers with bubbles; and $F_{diffusion}$ is the geometric mean of CH$_4$ fluxes in bubble free chambers.

Ebullition rate ($ER$; mL m$^{-2}$ h$^{-1}$), which reflected the volume rate of released of accumulated bubbles was calculated as (Eq. S3).

$$ER = \frac{V}{A_f \times t}$$

(S3)

where parameters of $V$, $A_f$, and $t$ are as given in Eq. (2).

[Figure]

[Figure]

**Figure S1.** Positive relationships between the ebullitive CH4 emission and (a) ebullition rates, (b) bubble CH4 concentrations in the NW transect

[Figure]

**Figure S2.** Seasonal variability in $CH_4$ emissions from the three reservoir areas

Note: The different letters marked in Fig. S2 indicated that the significant difference was found in the NE transects among the different seasons

[Figure]

**Figure S3.** Mean CH$_4$ emission flux at the each sampling point of different transects

[Figure]

[Figure]

[Figure]

**Figure S4.** Linear regressions between monthly mean water levels and monthly average CH₄ emissions in (a) NE, (b) SW, and (c) SE transects

[revised manuscript text omitted]

---

## Author Comment (AC2) · 15 Jul 2018

Reported greenhouse gas fluxes from reservoirs located in subtropical zone are still insufficient to understand regional carbon cycling. The manuscript titled "Seasonal and spatial variability of methane emissions form a subtropical reservoir in Eastern China" presents methane effluxes from a reservoir with an age of about 60 years, which is helpful to understand the topic. Although I would give it a positive response, a major revision should be made for being reviewed again. I hope some key environmental factors (wind speed, water velocity, water temperature, air temperature, air temperature, water depth of each site in different periods, and so on) should be introduced to elucidate their viewpoints instead of guessing. Response: Wind speed, water temperature, air temperature, and the air-water temperature difference were correlated with CH4 flux at each site. Results showed that weak relationships between wind speed, air-water temperature difference and CH4 flux were found (Tables 2, 3), and there were no relationships between air temperature (or water temperature) with CH4 flux. In addition, water levels were also correlated with CH4 flux (Figure S4), which account for 15-48% of CH4 flux variability. Based on these available data, some reasonable speculations were given in lines 298-305.

Specified comments: 1.Lines 125-137: Water depths should be presented for each sites. Response: Water depths of the sampling points were given in a new table (Table 1).

2.Lines 201-208: I don't think a para-curve model is suitable to fit gas concentrations in a chamber over time. The cited paper of Hutchinson and Livingston is about measuring gas exchange between soil and atmosphere. Pls see the following references. Xiao, S., Wang, C., Wikinson, R.J., Liu, D., Zhang, C., Xu, W., Yang, Z., Wang, Y., Lei, D., 2016. Theoretical model for diffusive greenhouse gas fluxes estimation across water-air interfaces measured with the static floating chamber method. Atmospheric Environment 137, 45-52. Tanka P. Kandel, Poul Eriklarke, Lars Elsgaard. Effect of chamber enclosure time on soil respiration flux: a comparison of linear and non-linear flux calculation methods. Atmospheric Environment 141, 245-254. Response: As the reviewer's suggestions, the linear regression was used to calculate the dc/dt instead of paea-curve model (line 186-188). Thus, the CH4 flux data of manuscript was changed because of the different regression methods (Result section).

3.Line 218: Are there some mistakes in the equation (2)? The denominators m, n and i may be 5, 13 and 3 respectively. Response: The denominators m, n and i are 5, 13 and 3, respectively. No error. But the sampling points in different transects were distinct in the 5 transects, so m=5 in the NE, SW, and SE transects, m=4 in the downstream river, and m=3 in the NW transect. The original equation 2 moved into the supplement

materials (Eq. S1) in the revised manuscript, according to the Review #1's suggestions.

4.Line 126-128: I think the natural physical grographical characteristics describes here is important. I think the changing hydrological situation may influence gas fluxes in the NW lake more or less. For example, can you correlate it with Lines of 315-328? Response: The relevant hydrological data were not measured during our sampling campaigns, so it was difficult to correlate it with CH4 flux. However, the data of mean monthly water levels were available to us, and average CH4 fluxes every month were correlated with their corresponding water levels (Seen Figure S4). From the results, water level fluctuation accounted for 15-48% of variability of CH4 fluxes.

5.Lines 354-370 This paragraph tries to explain "The seasonal variability of CH4 emission from the main body of Xin'anjiang Reservoir". However, I think they are feeble. I hope more environmental factors should be taken into considerations, such as wind speed, difference between the air temperature and water temperature, and so on. Response: According to the reviewer's suggestions, wind speed, air-water temperature difference were correlated with the CH4 fluxes in each transect (Table 2, Table 3). Wind speed accounted for 6.3-19% variability of CH4 fluxes in the reservoir main body (Table 2), and the air-water temperature difference accounted for 6-29% of CH4 fluxes variability (Table 3). Furthermore, seasonal dynamics of CH4 emission from the reservoir main body showed a similar pattern with the water level fluctuations in 2015 (Figure 1, Figure 5, Figure S4), and both of them had a high level in the second half of year. In addition, the thermocline depth and oxycline depth increased in Xin'anjiang Reservoir (called Qiandaohu Lake in Zhang et al. 2015) from July to Dec. and disappeared in Feb. and Mar. of next year. The stratification weakness periods (i.e., the second half of year) probably had an impact on CH4 oxidation and eventually CH4 emission at air-water interface, and the mechanism needed to be studied further (lines 293-305).

6.Lines 372-382: Reasons presented here for the high value of methane flux in the SW lake on August 1 are also weak. Figure 5 shows the methane flux in Feb. is also high, but the water level during the mon was low. Response: The 2 peaks on 1 August and

8 February were caused by the different reasons. The peak in 1 August was probably caused by the decomposition of vegetation in the littoral area. The average CH4 flux value decreased from the shallow area to the deep area (CH4 flux: SWP1, 0.47 mg m-2 h-1; SWP2, 0.33 mg m-2 h-1; SWP3, 0.098 mg m-2 h-1; SWP4, 0.1mg m-2 h-1; SWP5, 0.078mg m-2 h-1), and SWP1 and SWP2 was located in the shallow area (Table S4), and some submerged plants were distributed there. However, the CH4 peaks on 8 February was caused by the strong wind speed, and the average wind speed reached 8-10 m/s when the gas samples were collected in SWP2 and SWP4, with average measured CH4 fluxes of 0.23 mg CH4 m-2 s-1 and 0.20 mg CH4 m-2 s-1 respectively (lines 320-326).

7.English language needs a bigger improvement. For example: 1). Line 233: "from" should be "across". 2) Line 271: "individual measurements" should be "The flux of individual measurements." Response: English language has been polished by a professional company. The two errors were corrected in the revised manuscript.

―――――――――――――――